# A Criterion of Heat Transfer Deterioration for Supercritical Organic Fluids Flowing Upward and Its Heat Transfer Correlation

**Yung-Ming Li** [1], **Jane-Sunn Liaw** [2] **and Chi-Chuan Wang** [1,*]

[1] Department of Mechanical Engineering, National Chiao Tung University, Hsinchu 300, Taiwan; ms0466323@gmail.com

[2] Green Energy & Environment Research Laboratories, Industrial Technology Research Institute, Hsinchu 31040, Taiwan; jsliaw@itri.org.tw

[*] Correspondence: ccwang@nctu.edu.tw; Tel.: +886-3-5712121

**Abstract:** The main objective of this study was to develop the supercritical heat transfer correlation applicable for organic fluids when flowing upward in smooth tubes based on the available experimental data. The organic fluids contain R-22, R-134a, R-245fa and Ethanol and the associated heat transfer characteristics were compared with non-organic fluids like water and carbon-dioxide ($CO_2$). It was found that the limit heat flux may result in heat transfer deterioration (HTD) of organic fluid and the corresponding values are much smaller than water or $CO_2$. A new criterion to predict the HTD was developed and this criterion yields the best predictive ability against database. It was found that HTD occurs can be well described by the acceleration parameter evaluated at the wall condition rather than at bulk condition. For estimation of the supercritical heat transfer coefficient (HTC) for organic fluid, the present study proposes a new correlation with a physically based correction factor, which gives satisfactory predictions against the HTC of supercritical organic fluid. The new correlation can offer the smallest average deviation of 0.007 and standard deviation of 0.181 among the existing correlations.

**Keywords:** organic fluid; supercritical heat transfer; vertical tube; limit heat flux; heat transfer deterioration

## 1. Introduction

The cycling process of an Organic Rankine Cycle (ORC) is similar to Rankine cycle except the working fluid is organic. ORC can harvest low grade energy such as geothermal energy, solar energy, etc. [1]. However, the thermal efficiency for typical ORC is comparatively low. By changing the subcritical cycle to the trans-critical cycle, the efficiency of ORC can be improved because a better temperature can be reached in the evaporator, and higher exergy is attainable [2–4]. However, the estimation to heat transfer performance in the supercritical heating process is much more difficult than the subcritical heating process due to the immense variations of physical properties in the critical/pseudo-critical regions [3].

Figure 1 shows the enormous variations of properties for R-22 [5] at a certain temperature, called the pseudo-critical point. At a pseudo-critical point at a certain specified pressure above the critical point, the specific heat capacity may achieve to the maximum, and the density, viscosity, and thermal conductivity decrease remarkably. Yet, the variations become even more pronounced when the pressure is close to the critical point.

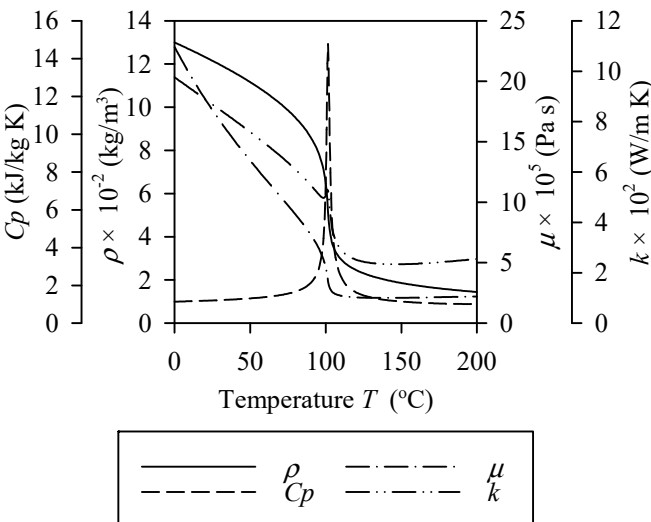

**Figure 1.** The enormous variations of physical properties for R-22 at a pressure of 5.5 MPa [5], where $\rho$ is density, $Cp$ is specific heat capacity, $\mu$ is viscosity, and $k$ is thermal conductivity, respectively.

The gigantic change of physical properties also results in huge change in heat transfer coefficients (HTC). Figure 2 shows the comparison of measured HTC against Dittus-Boelter correlation [5] for R-22. As depicted in the Figure, the correlation is applicable only at a lower heat flux (10 kW/m$^2$) or at a temperature that is some distance away from the pseudo-critical point. A further rise in heat flux to 20 kW/m$^2$ causes a significant over-prediction of the HTC, and it becomes more conspicuous with the heat flux. This phenomenon of reduction of HTC is called heat transfer deterioration (HTD) [6,7] and the lowest heat flux that leads to HTD is called limit heat flux (LHF) [5,8]. The phenomenon occurs due to the rapid change of density, which induces significant effects regarding buoyancy and acceleration [9–11]. In this regard, it is imperative to have an accurate correlation to predict the occurrence of HTD for supercritical fluid.

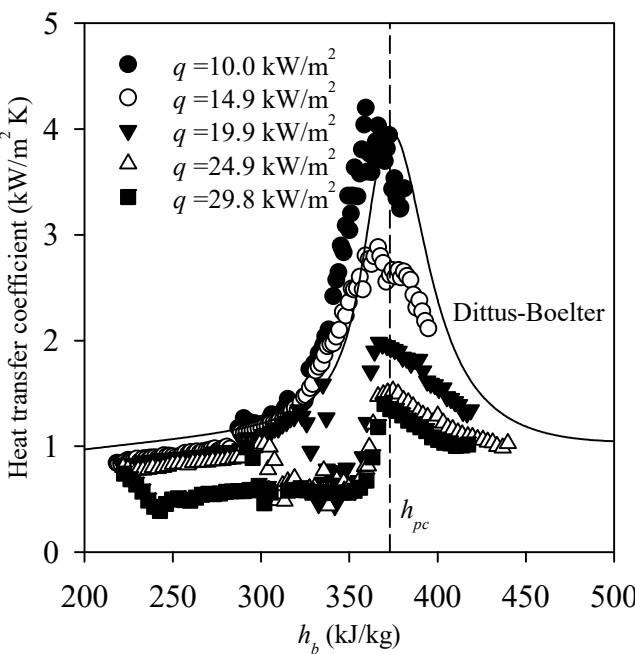

**Figure 2.** The comparison of experimental data for R-22 and Dittus-Boelter correlation [5], where $h_b$ and $h_{pc}$ signify bulk enthalpy and the enthalpy at the pseudo-critical point, respectively.

There have been many available correlations regarding HTD for inorganic fluids such as water and $CO_2$ and can be found in typical review articles (e.g., Jackson [11], Pioro et al. [12], and Yoo et al. [13]). However, there is few research concerning the heat transfer performance of organic fluids. The investigation for organic fluids in smooth tubes are listed on Table 1 [5,14–17]. Most investigations applied the correlations that were developed from water or carbon dioxide to predict the performance of organic fluids. Yamashita et al. [5] commented that the Watts and Chou [18] correlation developed from water gives the best agreement against the experimental data for R-22. Zhang et al. [16] suggested that the correlation proposed by Jackson [19] shows the best agreement against the experimental data for R-134a. He et al. [17] recommended two correlations that were originally developed from water proposed by Yamagata et al. [8] and Jackson [19] upon the R-245fa experimental data. On the other hand, the existing correlations developed for water or carbon dioxide all fail to predict the experimental data regarding the onset of HTD [16,17]. Several studies have developed new empirical correlations for organic fluids. For example, Kang and Chang [14] proposed a modified correlation based on the Jackson and Fewster correlation [20], and reported a mean deviation of 20% in predicting the Nusselt number for 94.6% of their R-134a experimental data. Zhang et al. [16] proposed a new simple modified correlation Dittus-Boelter correlation [21] by adding an acceleration parameter. The correlation is in good agreement with their R-134a test data. Notice that the foregoing correlations are only applicable for their test data only and no general heat transfer correlation for organic fluids yet.

**Table 1.** Investigations for performance of the heat transfer coefficient (HTC) and heat transfer deterioration (HTD) for organic fluids in smooth tube at supercritical region [5,14–17].

| Reference | Organic Fluid | Investigation of Limit Heat Flux | Recommended Correlation | New Correlation | Ability to Predict others' Data |
|---|---|---|---|---|---|
| Yamashita et al. [5] | R-22 | Yes | Watts and Chou [18] | No | No |
| Kang and Chang [14] | R-134a | No | No | Yes | Yes |
| Jiang et al. [15] | Ethanol | Yes | No | No | No |
| Jiang et al. [15] | R-22 | Yes | No | No | No |
| Zhang et al. [16] | R-134a | Yes | Jackson [19] | Yes | No |
| He et al. [17] | R-245fa | Yes | Yamagata et al. [8], Jackson [19] | No | No |

There were also some review papers regarding the heat transfer performance and the occurrence of HTD for supercritical water and carbon dioxide in the smooth tube [7,22,23]. Figure 3 compares the LHF against the mass flux of water, carbon dioxide, and the organic fluids investigated from the existing literature [5,14–17,23]. As shown in Figure 3, water contains the highest LHF among these fluids with the same mass flux, followed by the carbon dioxide, and lastly by the organic fluid. Comparing the criteria with the data of LHF, Yamagata's criterion [8], which is developed by water, shows good agreement only with water. Kim's criterion [24], which is developed based on carbon dioxide, is also only applicable for carbon dioxide. As can be clearly seen, both criteria over-predict the LHF of organic fluids, especially Yamagata's criterion.

Moreover, the quantitative comparison with experimental data of LHF and the existing criteria in terms of polynomial form against the mass flux is listed in Table 2. The average deviation (*AD*) and standard deviation (*SD*) are defined [10] below:

$$AD = \frac{1}{N}\sum_{i=1}^{N}\frac{2(\mathrm{LHF}_C - \mathrm{LHF}_M)_i}{(\mathrm{LHF}_C + \mathrm{LHF}_M)_i}, \tag{1}$$

$$SD = \left[\frac{1}{N-1}\sum_{i=1}^{N}\left(AD - \frac{2(\mathrm{LHF}_C - \mathrm{LHF}_M)_i}{(\mathrm{LHF}_C + \mathrm{LHF}_M)_i}\right)\right]^{1/2}. \tag{2}$$

The criteria developed by water and carbon dioxide were discussed in the review by Huang et al. [23]. The result shows that the criteria can provide good predictions only for its database,

extrapolations to other working fluids are normally futile. The criteria developed for water (e.g., Yin et al. [25], Yamagata et al. [8], Styrikovich et al. [26], and Mokry et al. [27]) could provide an absolute average deviation of below 0.6 and a standard deviation of less than 0.3. The criteria developed for carbon dioxide such as Kim et al. [24] could provide an absolute average deviation less than 0.3 with a standard deviation being lower than 0.6. However, it also reveals that these criteria over-predict the data of organic fluids, especially for those criteria developed for water. The average deviations are at least 0.5. Based on the aforementioned reviews, it was concluded that no appropriate criteria of LHF is applicable yet.

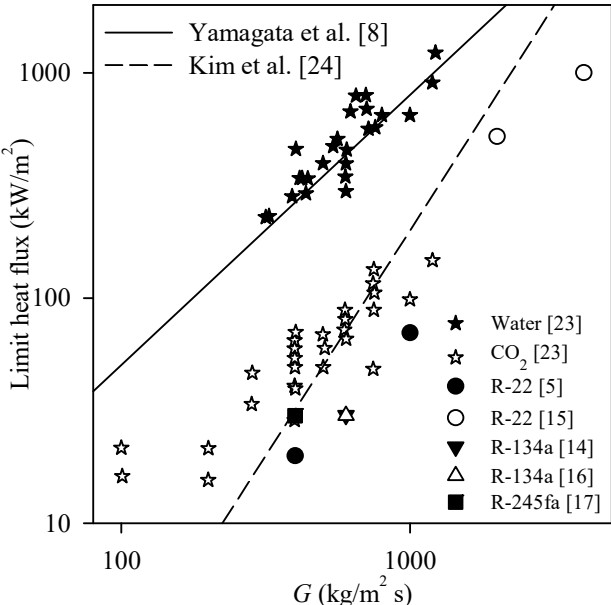

**Figure 3.** Comparisons of the limit heat flux (LHF) between correlations and experimental data against the mass flux for water, carbon dioxide and the organic fluids [5,14–17,23].

**Table 2.** The criteria and predicted the average deviation (*AD*) and standard deviation (*SD*) of the LHF of water and carbon dioxide [23].

| References | Criterion | For | | Water | Carbon Dioxide | Organic Fluids |
|---|---|---|---|---|---|---|
| Yin et al. [25] | $LHF = \frac{1}{2.16}G$ | Water | *AD* | −0.535 | 1.138 | 1.293 |
| | | | *SD* | 0.198 | 0.181 | 0.448 |
| Yamagata et al. [8] | $LHF = 0.2G^{1.2}$ | Water | *AD* | −0.119 | 1.358 | 1.551 |
| | | | *SD* | 0.216 | 0.180 | 0.243 |
| Styrikovich et al. [26] | $LHF = 0.58G$ | Water | *AD* | −0.323 | 1.279 | 1.407 |
| | | | *SD* | 0.208 | 0.159 | 0.394 |
| Kim et al. [24] | $LHF = 0.0002G^2$ | $CO_2$ | *AD* | −1.470 | −0.299 | 0.584 |
| | | | *SD* | 0.189 | 0.581 | 0.387 |
| Mokry et al. [27] | $LHF = -58.97 + 0.745G$ | Water | *AD* | −0.232 | 1.219 | 1.471 |
| | | | *SD* | 0.211 | 0.417 | 0.324 |

In view of the foregoing review, despite the fact that there were many studies concerning the predictions about the limit heat flux and the occurrence of the HTD in the literature, there were simply no general criteria and correlations for organic fluids. Hence, the objective of this study was to tailor and bridge the gap in developing the associated criteria and correlations for engineering elaborations. By collecting and analyzing exiting experimental data in the literature, the main objective of this study was to propose a new criterion to describe the heat transfer deterioration of supercritical organic fluids. Yet, a corresponding heat transfer correlation based on the experimental data was also developed that will be shown subsequently to offer a superior predictive ability to the existing correlations.

## 2. Materials and Methods

### 2.1. Data Analysis

The related studies [5,14–17] in association with organic fluids are tabulated in Table 3 and all the data points are included as the database for developing the criteria and correlations. Yamashita et al. [5] carried out the experiment for supercritical R-22 flowing upward in the vertical tube. The hydraulic diameter is 4.4 mm with the heated length of 2 m. Their range of operating condition included the pressure of 5.5 MPa, the mass flux of 400–2000 kg/m$^2$·s, and wall heat flux of 10–170 kW/m$^2$. Kang and Chang [14] carried out the experiment for supercritical R-134a flowing upward in the vertical tube. The diameter of the test tube is 9.4 mm with a heated length of 2 m. The operating conditions were with the inlet pressure of 4.1, 4.3, and 4.5 MPa, respectively, while the wall heat flux is from 10 to 160 kW/m$^2$·s and the mass flux ranges from 600 to 2000 kg/m$^2$·s. Jiang et al. [15] conducted an experiment of supercritical R-22 and ethanol. The diameter of the tube was 1.004 mm. Their wall heat flux ranged from 110 to 1800 kW/m$^2$, while the fluid inlet Reynold number ranged from 3500 to 24000, and the pressure spanned from 5.5 MPa to 10 MPa. Zhang et al. [16] performed the experiment for the supercritical R-134a flowing upward in the vertical tube. The hydraulic diameter was 7.6 mm with the heated length of 2.3 m. Their operating conditions included the pressure of 4.3–4.7 MPa, the mass flux of 600–2500 kg/m$^2$ s, and the wall heat flux of 20–180 kW/m$^2$. He et al. [17] carried out the experiment for supercritical R-245fa flowing upward in the vertical tube. The hydraulic diameter is 4 mm with the heated length of 1.04 m. Operating conditions included the pressure of 4–5 MPa, the mass flux of 400–800 kg/m$^2$·s, and wall heat flux of 15–100 kW/m$^2$.

Prior studies addressed some important factors on the thermofluids characteristics such as fluid flow condition, flow direction, geometry, etc. Yet, it was found that the buoyancy and acceleration effects impose severe effect on the heat transfer performance especially for the upward flowing conditions. Hence, this study stresses only for smooth tubes subject to upward flow conditions. To sum up, the fluid investigated includes R-22, R-134a, R-245fa, and ethanol. The total number of data used to develop the correlation is 4260. The diameter ranges from 1 to 10 mm, and the corresponding mass flux and heat flux ranges from 400–4000 kg/m$^2$·s, and 10–1800 kW/m$^2$, respectively.

**Table 3.** The experimental data from the literatures [5,14–17].

| References | Fluid | N | L (m) | D (mm) | P (MPa) | G (kg/m$^2$·s) | q (kW/m$^2$) | $T_b$ (°C) | $T_w$ (°C) |
|---|---|---|---|---|---|---|---|---|---|
| Yamashita et al. [5] | R-22 | 927 | 2 | 4.4 | 5.5 | 400–2000 | 10–170 | 12–120 | 26–154 |
| Kang and Chang [14] | R-134a | 560 | 2 | 9.4 | 4.1–4.5 | 600–2000 | 10–160 | 49–113 | 59–188 |
| Jiang et al. [15] | Ethanol | 262 | 0.152 | 1.004 | 5.5–10 | 2000–4000 | 110–1800 | 23–110 | 46–326 |
| Jiang et al. [15] | R-22 | 345 | 0.152 | 1.004 | 5.5–10 | 2000–4000 | 110–1800 | 22–183 | 41–384 |
| Zhang et al. [16] | R-134a | 530 | 2.3 | 7.6 | 4.3–4.7 | 600–2500 | 20–180 | 74–107 | 86–182 |
| He et al. [17] | R-245fa | 1636 | 1.04 | 4 | 4–5 | 400–800 | 15–100 | 100–195 | 121–234 |

### 2.2. Data Reduction

Since part of the database only discloses the bulk enthalpy and wall temperature, the corresponding bulk temperature and corresponding physical properties were evaluated with the help of REFPROP ver. 8 [28] from the prescribed pressures and the bulk enthalpies. To further investigate the heat transfer performance, the heat transfer coefficient (HTC) was used and analyzed in the following:

$$\text{HTC} = \frac{q}{T_w - T_b},\tag{3}$$

where $q$ represents the heat flux and $T_w$ and $T_b$ represent wall and bulk temperature, respectively. To facilitate detailed comparisons amid the proposed and some existing correlations, the HTC is further termed into dimensionless Nusselt number:

$$Nu = \frac{\text{HTC} \times D}{k}, \tag{4}$$

where $D$ represents the diameter, and $k$ is thermal conductivity. The collected experimental data is further analyzed through Matlab 2018b. Quantitative comparisons with experimental data of Nusselt number and the existing correlation are termed as the average deviation $AD$ and the standard deviation $SD$ [10] in the following:

$$AD = \frac{1}{N} \sum_{i=1}^{N} \frac{2(Nu_C - Nu_M)_i}{(Nu_C + Nu_M)_i}, \tag{5}$$

$$SD = \left[ \frac{1}{N-1} \sum_{i=1}^{N} \left( AD - \frac{2(Nu_C - Nu_M)_i}{(Nu_C + Nu_M)_i} \right) \right]^{1/2}. \tag{6}$$

## 3. Results and Discussion

### 3.1. The Criterion of the Lowest Heat Flux for Heat Transfer Deterioration

Figure 4 shows the Nusselt number versus the enthalpy at different conditions from Yamashita et al. [5]. As addressed in prior studies, the effect of heat flux occurs only when buoyancy and acceleration effects is in control and this influence is accentuated nearby pseudo-critical regions. For a mass flowrate of 400 kg/m$^2$·s, the Nusselt number increases with enthalpy and then level-off appreciably when it reaches the vicinity of pseudo-critical enthalpy. The peak of Nusselt number becomes less profound when raising the heat flux. However, when heat flux achieves to a threshold value such as 30 kW/m$^2$ (for $G$ = 400 kg/m$^2$·s), the Nusselt number suddenly drops before the bulk temperature reaches the pseudo-critical temperature. In essence, HTD occurs accordingly.

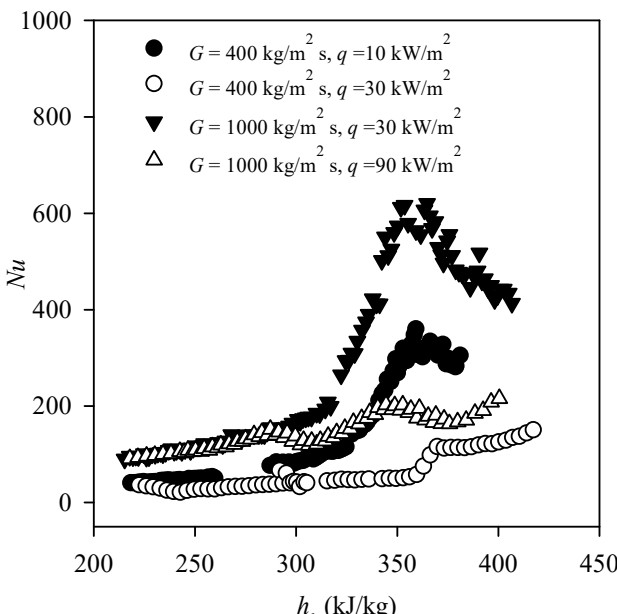

**Figure 4.** Nusselt number vs. the enthalpy subject to the heat flux and mass flux [5].

At a mass flow rate of 1000 kg/m$^2$·s, the trend of HTD was similar to low mass flow rate of 400 kg/m$^2$·s. However, the Nusselt number of a mass flow rate of 1000 kg/m$^2$·s was higher than the one of 400 kg/m$^2$·s at lower enthalpy region for its much higher Reynolds number. Interestingly, at the heat flux at 30 kW/ m$^2$, HTD did not happen at a mass flow rate of 1000 kg/ m$^2$·s as that of $G$ = 400 kg/m$^2$·s. Instead, HTD occurred at a higher heat flux such as 90 kW/m$^2$. It reveals the lowest heat flux leading the HTD, which is called limit heat flux (LHF) [5,8], which depends on mass flux.

As aforementioned, there were no appropriate criteria of LHF using polynomial forms for organic fluid. However, Cheng et al. [10] proposed a different polynomial form as the criterion of LHF (Equation (7)). The authors claimed that the property of fluid is included in the criterion:

$$\frac{\text{LHF}}{G} = 1.354 \cdot 10^{-3} \frac{Cp_{PC}}{\beta_{PC}}, \tag{7}$$

where $Cp_{PC}$ and $\beta_{PC}$ is specific heat capacity and thermal expansion coefficient at pseudo-critical condition, respectively.

Figure 5 shows a good agreement between the experimental LHF of water and carbon dioxide and the criterion of Cheng et al. [10]. Moreover, the criterion of Cheng et al. [10] contained the effect of pressure in comparison with Yamagata's criterion [8]. Although the criterion by Cheng et al. [10] cannot predict the HTD of the organic fluids well, it revealed that the ratio of $\beta$ to $Cp$ may affect the occurrence of HTD. The ratios are listed in Table 4, and it was found that the ratios for the organic fluids except ethanol ranging from $13 \times 10^{-6}$ to $16 \times 10^{-6}$ kg/J. The values were much higher than $1.79 \times 10^{-6}$ kg/J of water and $8.80 \times 10^{-6}$ kg/J of carbon dioxide at a near critical pressure. The higher ratios lead to the occurrence of HTD at a quite low heat flux due to the higher level of thermal expansion or the higher variety of density for given a heat flux or mass flux. The magnitude of the ratios of $\beta$ to $Cp$ can explain the LHF of organic fluids that is lower than water and carbon dioxide at a given mass flux.

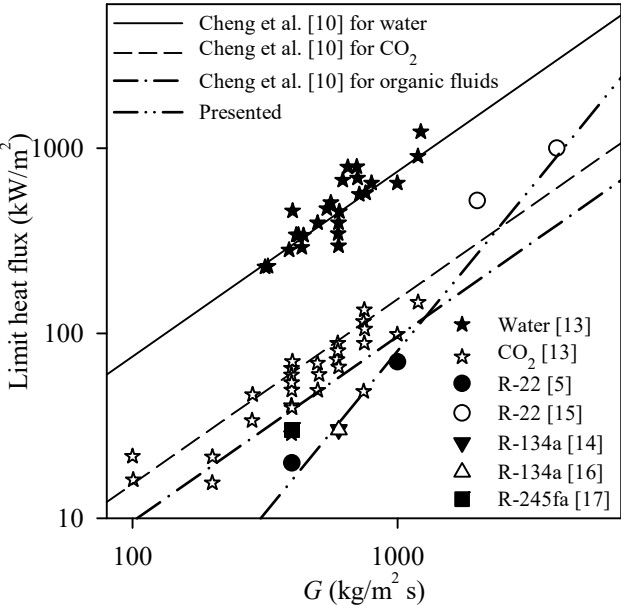

**Figure 5.** Comparison between experimental LHF and the criterion proposed by Cheng et al. [10].

Until now, there is no criterion for organic fluid. It is difficult to use Cheng et al.'s form to develop the criterion for HTD of organic fluids. As a result, the study performed regression upon the experiment data and proposes a new criterion for the organic fluids as shown in the following:

$$\text{LHF} = 4.5 \times 10^{-4} G^{1.75}. \tag{8}$$

The correlation is valid for ratios ranging from $13 \times 10^{-6}$ to $16 \times 10^{-6}$ kg/J and the fluid of R-22, R-134a, and R-245fa. The quantitative comparison amid the experimental data and the criteria proposed by Cheng et al. [10] and the proposed criterion is shown as Table 5. It also reveals a good agreement between the experimental LHF of water and carbon dioxide and the criterion of Cheng et al [10]. However, their correlation showed a deviation of 0.650 for organic fluids, which is much higher than the standard deviation of 0.336 upon the proposed new criterion. Moreover, the absolute average deviation of 0.232

for organic fluids is lower than the deviation provided by the existing criteria form Table 2. In conclusion, the proposed new criterion of LHF for organic fluids shows the best predictive ability.

**Table 4.** The ratio of $\beta$ to $Cp$ for water, carbon dioxide, and organic fluids.

| Fluid | $P$ (MPa) | $\beta_{pc}/Cp_{pc} \times 10^6$ (kg/J) |
|---|---|---|
| Water | 22.6 | 1.79 |
| $CO_2$ | 7.5 | 8.80 |
| R-22 | 5.5 | 13.43 |
| R-134a | 4.3 | 15.27 |
| R-245fa | 4 | 15.77 |
| | 4.5 | 13.23 |

**Table 5.** The criteria of the LHF of water and carbon dioxide [23].

| Author | Criterion | For | | Water | Carbon Dioxide | Organic Fluids |
|---|---|---|---|---|---|---|
| Cheng et al. [10] | $\frac{\text{LHF}}{G} = 1.354 \cdot 10^{-3}\frac{Cp_{PC}}{\beta_{PC}}$ | Water | AD | −0.069 | 0.199 | 0.111 |
| | | | SD | 0.213 | 0.271 | 0.650 |
| This study | $\text{LHF} = 4.5 \times 10^{-4}G^{1.75}$ | Organic | AD | - | - | −0.232 |
| | | | SD | - | - | 0.336 |

## 3.2. The onset of Laminarization and HTD

The occurrence of HTD is related to acceleration [9,11]. When the effect of acceleration is increased, the turbulence might be suppressed and the heat transfer performance decreases. The dimensionless form of acceleration factor is proposed by McEligot et al. [9] as $Kv$ (acceleration factor):

$$Kv = \frac{\nu_b}{u_b^2}\frac{\partial u_b}{\partial x} = -\frac{D}{\text{Re}}\beta_p\frac{dp}{dx} + \frac{4q_wd\beta}{\text{Re}^2\mu_bCp}. \tag{9}$$

McEligot et al. [9] suggested that the flow changes from turbulent to laminar when $Kv$ is higher than $3 \times 10^{-6}$. However, Jiang et al. [15] indicated that $Kv$ for R-22 is far less than the threshold value proposed by McEilgot et al. [9]. On the other hand, Cheng et al. [10] proposed a simpler parameter $\pi_A$ to describe the acceleration effect, which is defined as:

$$\pi_A = \frac{D}{\rho}\frac{\partial\rho}{\partial x} = \frac{q\beta}{GCp}. \tag{10}$$

Obviously, $Kv$ is more complex than $\pi_A$. Hence, using $\pi_A$ is easier to implement from an engineering perspective and will be adopted in the following developments.

Figures 6–8 depict the corresponding Nusselt number and acceleration factor $\pi_A$ at both bulk and wall condition versus enthalpy for R-22, R-134a, and R-245fa. It was found that the peak of $\pi_A$ is increased with a rise of heat flux. When compared with Nusselt number and $\pi_A$ at bulk condition, the Nusselt number will decrease first, followed by an increase against the enthalpy when $\pi_A$ passes through a certain value against the enthalpy. This is the basic cause for deterioration of heat transfer performance. Manipulation and combination of the criterion of LHF by Equation (8) and $\pi_A$ yields the corresponding threshold value:

$$\pi_{A,th} = \frac{\text{LHF} \times \beta_{pc}}{GCp_{pc}}. \tag{11}$$

The value shown as dash line in Figure 6 indicates that HTD does occur in the condition when the peak of $\pi_A$ is beyond the threshold. However, it cannot predict the location where HTD occurs in association with $\pi_A$ at bulk condition. Hence, $\pi_A$ at the wall condition should be also taken in account.

By comparing Nusselt number and $\pi_A$ at wall condition, it is found that HTD occurs in the vicinity even after $\pi_A$ at wall condition approaches the threshold value. This correlation can predict the onset of HTD due to laminarization [9,11].

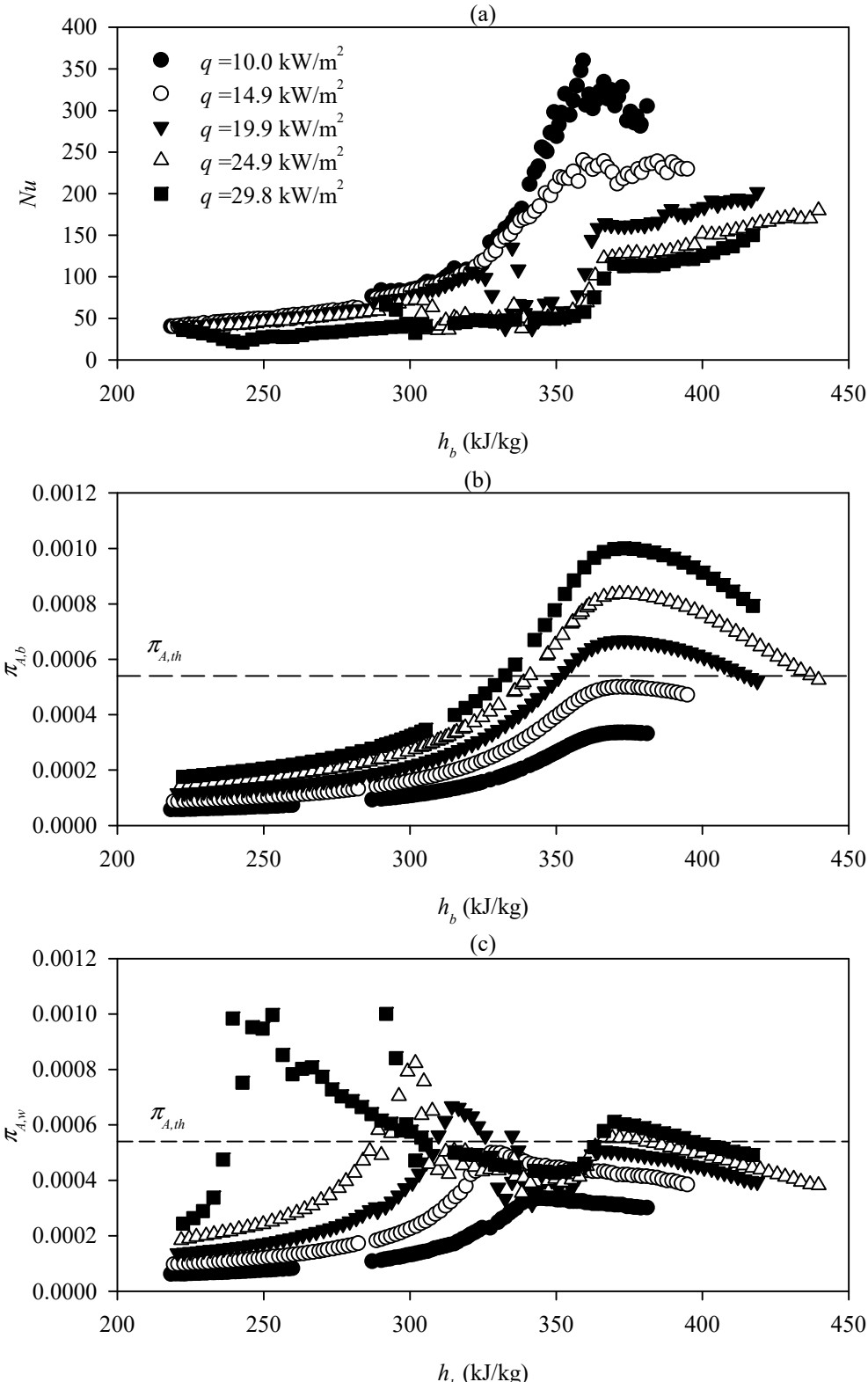

**Figure 6.** (**a**) Nusselt number, (**b**) $\pi_A$ at both bulk and (**c**) wall condition vs. enthalpy for R-22 at mass flux of 400 kg/ m$^2$·s subject to heat fluxes [5].

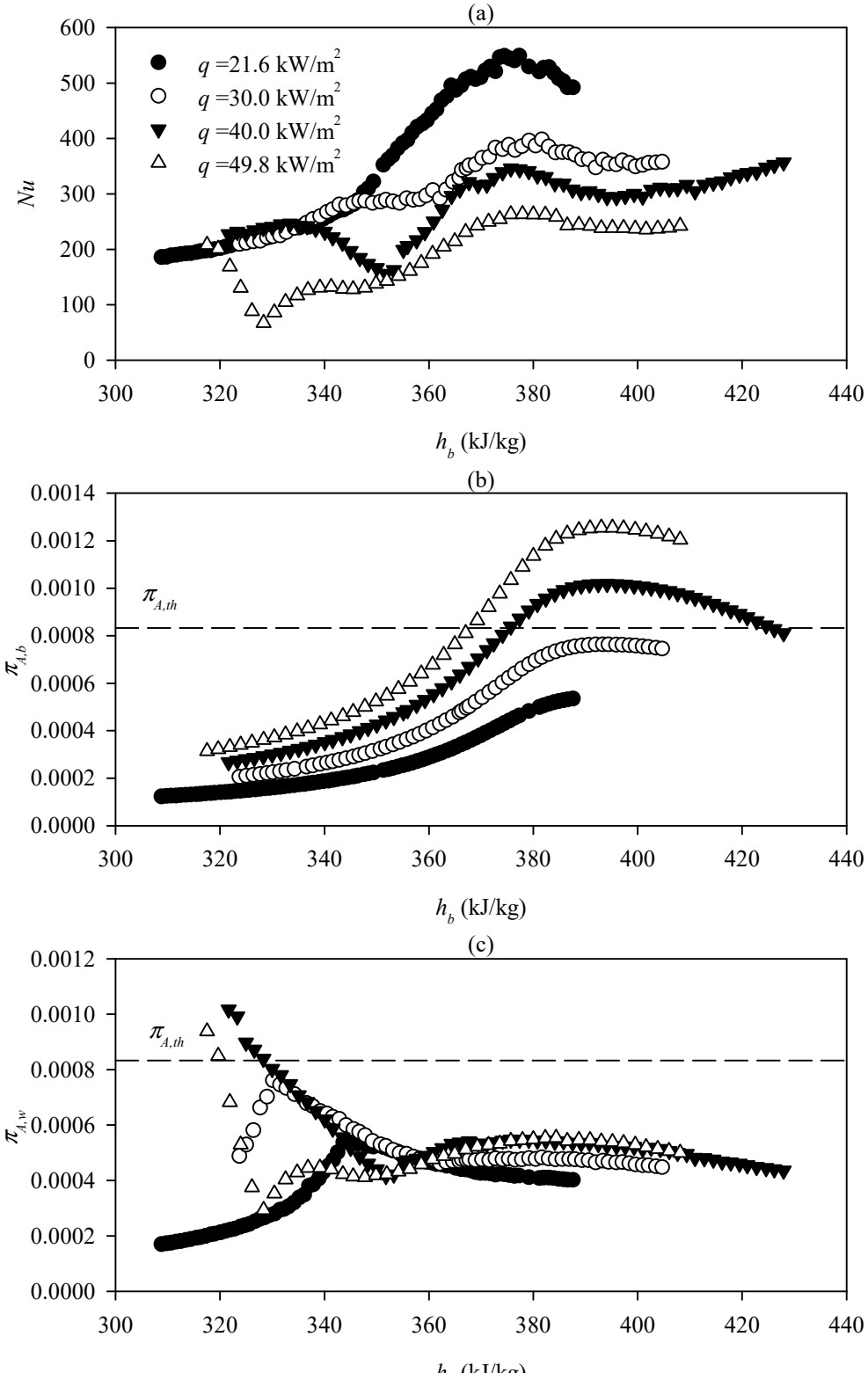

**Figure 7.** (**a**) Nusselt number, (**b**) $\pi_A$ at both bulk and (**c**) wall condition vs. enthalpy for R-134a at mass flux of 600 kg/ m$^2$·s subject to heat fluxes [16].

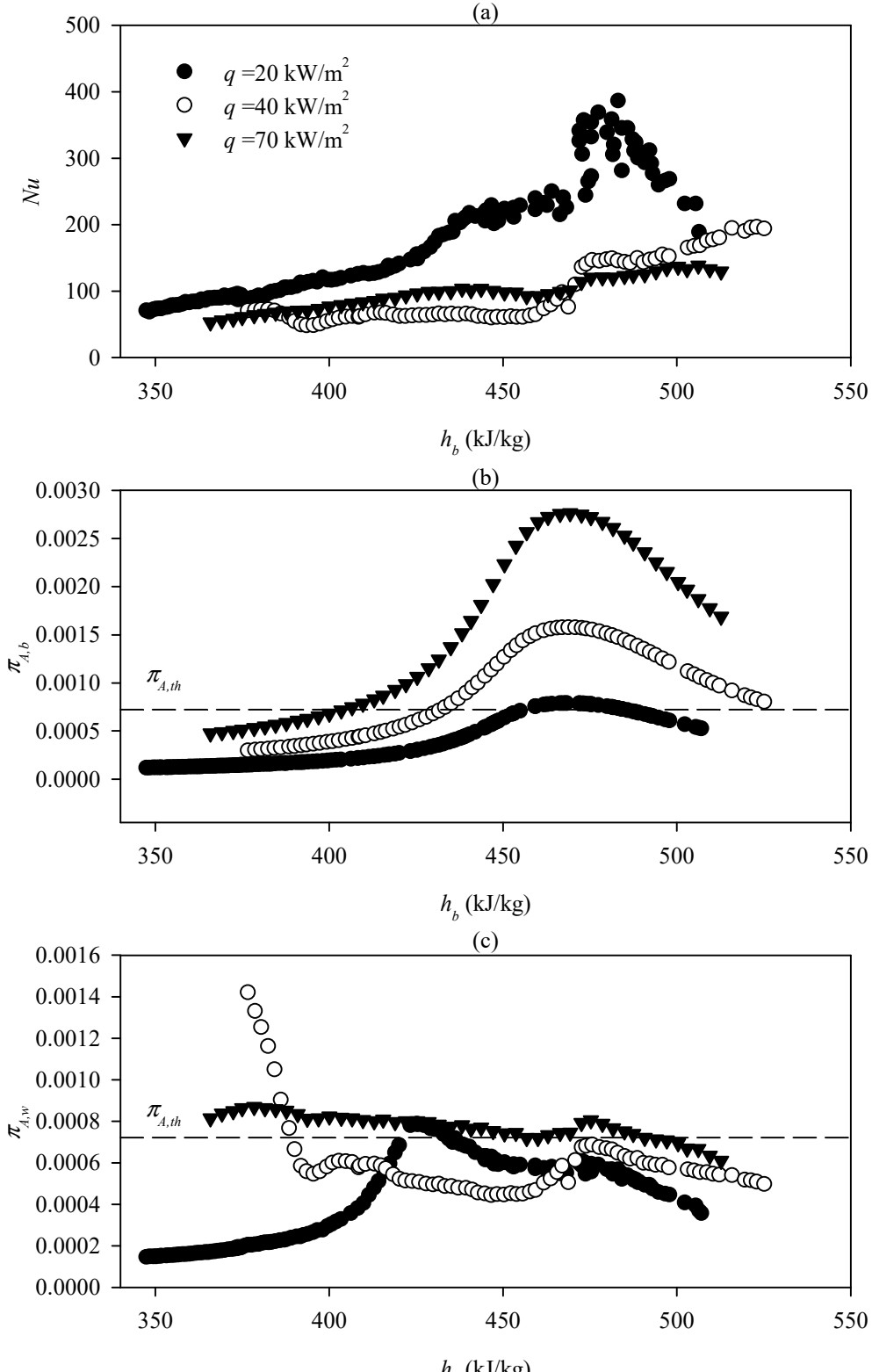

**Figure 8.** (**a**) Nusselt number, (**b**) $\pi_A$ at both bulk and (**c**) wall condition vs. enthalpy for R-245fa at mass flux of 400 kg/ m$^2$·s subject to heat fluxes [17].

## 3.3. The Correlations of the Heat Transfer

### 3.3.1. The Comparison Between the Experimental Data and the Existing Correlations

Table 6 lists some correlations that are employed for further comparison against the database from the literature.

**Table 6.** Some correlations used for comparisons against the experimental data subject to supercritical conditions.

| Authors | Correlation | Fluid |
|---|---|---|
| Dittus-Boelter [21] | $Nu = 0.023Re_b^{0.8}Pr_b^{0.4}$ $Re_b = GD/\mu_b$ $Pr_b = \mu_b Cp_b/k_b$ | Water |
| Krasnoshchekov et al. [29] | $Nu = \dfrac{(\xi/8)Re_b\overline{Pr}}{12.7\sqrt{\xi/8}\left(\overline{Pr}^{2/3}-1\right)+1.07}\left(\rho_w/\rho_b\right)^{0.3}\left(\overline{Cp}/Cp_b\right)^n$ $\xi = (1.82\log_{10}Re_b - 1.64)^{-2}$ $\overline{Pr} = \mu_b\overline{Cp}/k_b$ $\overline{Cp} = (h_w - h_b)/(T_w - T_b)$ $n = \begin{cases} 0.4 & for\ T_w/T_{pc} \le 1\ or\ 1.2 \le T_b/T_{pc} \\ n_1 = 0.22 + 0.18\left(T_w/T_{pc}\right) & for\ 1 \le T_w/T_{pc} \le 2.5 \\ n_1 + (5n_1 - 2)\left[1 - \left(T_b/T_{pc}\right)\right] & for\ 1 \le T_b/T_{pc} \le 1.2 \end{cases}$ | Water, Carbon dioxide |
| Yamagata et al. [8] | $Nu = 0.0135Re_b^{0.85}Pr_b^{0.8}CF$ $CF = \begin{cases} 1 & for\ E > 1 \\ 0.67Pr_{pc}^{-0.05}\left(\overline{Cp}/Cp_b\right)^{n_1} & for\ 0 \le E \le 1 \\ \left(\overline{Cp}/Cp_b\right)^{n_2} & for\ Gr^* < 0 \end{cases}$ $E = \left(T_{pc} - T_b\right)/\left(T_w - T_b\right)$ $n_1 = -0.77\left(1 + 1/Pr_{pc}\right) + 1.49$ $n_2 = 1.44\left(1 + 1/Pr_{pc}\right) - 0.53$ | Water |
| Jackson and Fewster [20] | $Nu = 0.0183Re_b^{0.82}\overline{Pr}^{0.5}\left(\rho_w/\rho_b\right)^{0.3}$ | Water |
| Watts and Chou [18] | $Nu = 0.021Re_b^{0.8}\overline{Pr}^{0.55}\left(\rho_w/\rho_b\right)^{0.35}CF(Gr^*)$ $CF(Gr^*) = \begin{cases} 1 & for\ Gr^* < 10^{-5} \\ (1 - 3000Gr^*)^{0.295} & for\ 10^{-5} \le Gr^* \le 10^{-4} \\ (7000Gr^*)^{0.295} & for\ Gr^* > 10^{-4} \end{cases}$ $Gr^* = \dfrac{\overline{Gr}}{Re_b^{2.7}\overline{Pr}^{0.5}}$ $\overline{Gr} = \rho_b(\rho_b - \overline{\rho})gD^3/\mu_b^2$ $\overline{\rho} = \dfrac{1}{(T_w - T_b)}\int_{T_b}^{T_w}\rho(T)dT$ | Water |
| Jackson [19] | $Nu = 0.0183Re_b^{0.82}Pr_b^{0.5}\left(\rho_w/\rho_b\right)^{0.3}\left(\overline{Cp}/Cp_b\right)^n$ $n = \begin{cases} 0.4 & for\ T_b < T_w < T_{pc}\ or\ 1.2T_{pc} < T_b < T_w \\ 0.4 + 0.2\left(T_w/T_{pc} - 1\right) & for\ T_b < T_{pc} < T_w \\ 0.4 + 0.2\left(T_w/T_{pc} - 1\right)\left[1 - 5\left(T_b/T_{pc} - 1\right)\right] & for\ T_{pc} \le T_b \le 1.2T_{pc}\ and\ T_b < T_w \end{cases}$ | Water |
| Kang and Chang [14] | $Nu = 0.0244Re_b^{0.762}\overline{Pr}^{0.552}\left(\rho_w/\rho_b\right)^{0.293}$ | R-134a |
| Zhang et al. [16] | $Nu = 0.023Re_b^{0.8}Pr_b^{0.4}CF$ $CF = \min(F_1, F_2)$ $F_1 = 1.0 + 1936\pi_A^{1.059}$ $F_2 = -5.19 - 0.817\ln\pi_A$ $\pi_A = \dfrac{q\beta_b}{GCp_b}$ | R-134a |

Table 7 shows the comparisons between the experimental data and the existing correlations. The average deviation and standard deviation are defined in Eqs. (5-6) [10], which are used to compare the experimental data.

**Table 7.** Comparisons between the experimental data and the correlations.

| Correlation | | R-22 | R-22 | R-134a | R-134a | R-245fa | Ethanol | Total |
|---|---|---|---|---|---|---|---|---|
| | | Yamashita et al. [5] | Jiang et al. [15] | Kang and Chang [14] | Zhang et al. [16] | He et al. [17] | Jiang et al. [15] | |
| Dittus-Boelter [21] | N | 927 | 345 | 560 | 530 | 1636 | 262 | 4260 |
| | AD | 0.252 | 0.298 | 0.358 | 0.048 | −0.164 | −0.071 | 0.065 |
| | SD | 0.342 | 0.556 | 0.382 | 0.390 | 0.366 | 0.227 | 0.433 |
| Krasnoshchekov et al. [29] | N | 927 | 345 | 560 | 530 | 1636 | 262 | 4260 |
| | AD | 0.165 | −0.062 | 0.125 | 0.093 | −0.226 | 0.044 | −0.025 |
| | SD | 0.297 | 0.586 | 0.340 | 0.323 | 0.278 | 0.216 | 0.369 |
| Yamagata et al. [8] | N | 927 | 345 | 560 | 530 | 1636 | 262 | 4260 |
| | AD | 0.439 | 0.252 | 0.505 | 0.367 | 0.061 | 0.780 | 0.300 |
| | SD | 0.175 | 0.462 | 0.150 | 0.099 | 0.287 | 0.171 | 0.329 |
| Jackson and Fewster [20] | N | 927 | 345 | 560 | 530 | 1636 | 262 | 4260 |
| | AD | 0.207 | 0.050 | 0.226 | 0.063 | −0.194 | 0.143 | 0.021 |
| | SD | 0.184 | 0.467 | 0.157 | 0.112 | 0.241 | 0.202 | 0.294 |
| Watts and Chou [18] | N | 927 | 345 | 560 | 530 | 1636 | 262 | 4260 |
| | AD | 0.165 | −0.026 | 0.162 | 0.012 | −0.228 | 0.232 | −0.017 |
| | SD | 0.169 | 0.477 | 0.158 | 0.105 | 0.232 | 0.195 | 0.291 |
| Jackson [19] | N | 927 | 345 | 560 | 530 | 1636 | 262 | 4260 |
| | AD | 0.209 | 0.045 | 0.237 | 0.047 | −0.195 | 0.130 | 0.019 |
| | SD | 0.188 | 0.468 | 0.183 | 0.142 | 0.246 | 0.200 | 0.301 |
| Kang and Chang [14] | N | 927 | 345 | 560 | 530 | 1636 | 262 | 4260 |
| | AD | 0.068 | 0.014 | 0.117 | −0.040 | −0.311 | 0.114 | −0.086 |
| | SD | 0.272 | 0.468 | 0.172 | 0.174 | 0.310 | 0.225 | 0.339 |
| Zhang et al. [16] | N | 927 | 345 | 560 | 530 | 1636 | 262 | 4260 |
| | AD | 0.319 | -0.554 | 0.261 | 0.080 | −0.372 | 0.027 | −0.072 |
| | SD | 0.246 | 1.201 | 0.380 | 0.195 | 0.532 | 0.226 | 0.608 |

For the data of R-22, Kang and Chang's correlation [14] provides the smallest average deviation of 0.068 and 0.014 against the data of Yamashita et al. [5] and Jiang et al. [15], respectively. The correlations of Watts and Chou [18] and Yamagata et al. [8] provide the smallest standard deviation of 0.169 and 0.462 against the database of Yamashita et al. [5] and Jiang et al. [15], respectively.

For the data of R-134a, Kang and Chang's correlation [14] also provides the smallest average deviation of 0.117 and −0.040 against the data of both Kang and Chang [14] and Zhang et al. [16], respectively. The correlation of Yamagata et al. [8] provides the smallest standard deviation 0.150 and 0.099 against the database from Kang and Chang [14] and Zhang et al. [16], respectively.

For the data of R-245fa, Yamagata's correlation [8] provides the smallest average deviation of 0.061, and Watts and Chou's correlation [18] provide the smallest standard deviation.

For the data of ethanol from Jiang et al. [15], Zhang's correlation [16] offers the smallest average value, and Yamagata's correlation [8] provides the smallest standard deviation. For all data, Watts and Chou's correlation [18] can provide the smallest average deviation (−0.017) and standard deviation (0.291) among these correlations.

Moreover, the performance of empirical correlation for heat transfer to supercritical organic fluid is listed Table 8. It reveals that Kang and Chang correlation [14] provides the predictive ability against the R-22 and R-134a data. Watts and Chou's correlation [18] provides the best agreement against the R-245fa data. Zhang's correlation [16] offers the best agreement against the ethanol data. Obviously, the best agreement is obtained if the correlation has the smallest either average or standard deviation. It implies that there are no existing correlations that are able to provide satisfactory agreement against all organic data. In fact, Watts and Chou's correlation [18] shows the best overall predictive ability in which 73% of 4260 data falls within the ±30% span.

**Table 8.** Comparisons of the predictive ability of heat transfer coefficient at supercritical states between the existing correlations and the test data for organic fluids.

| Correlation | | R-22 | R-134a | R-245fa | Ethanol | Total |
|---|---|---|---|---|---|---|
| Dittus-Boelter [21] | $N$ | 1272 | 1090 | 1636 | 262 | 4260 |
| | ±20% | 42 | 48 | 25 | 55 | 37 |
| | ±30% | 55 | 61 | 52 | 87 | 57 |
| Krasnoshchekov et al. [29] | $N$ | 1272 | 1090 | 1636 | 262 | 4260 |
| | ±20% | 30 | 32 | 50 | 70 | 40 |
| | ±30% | 55 | 48 | 64 | 82 | 58 |
| Yamagata et al. [8] | $N$ | 1272 | 1090 | 1636 | 262 | 4260 |
| | ±20% | 11 | 0 | 47 | 0 | 22 |
| | ±30% | 20 | 7 | 67 | 0 | 34 |
| Jackson and Fewster [20] | $N$ | 1272 | 1090 | 1636 | 262 | 4260 |
| | ±20% | 43 | 66 | 48 | 62 | 51 |
| | ±30% | 69 | 76 | 68 | 72 | 70 |
| Watts and Chou [18] | $N$ | 1272 | 1090 | 1636 | 262 | 4260 |
| | ±20% | 52 | 76 | 47 | 47 | 55 |
| | ±30% | 72 | 86 | 69 | 59 | 73 |
| Jackson [19] | $N$ | 1272 | 1090 | 1636 | 262 | 4260 |
| | ±20% | 45 | 63 | 45 | 63 | 50 |
| | ±30% | 68 | 73 | 68 | 73 | 69 |
| Kang and Chang [14] | $N$ | 1272 | 1090 | 1636 | 262 | 4260 |
| | ±20% | 63 | 78 | 18 | 63 | 50 |
| | ±30% | 74 | 89 | 43 | 77 | 66 |
| Zhang et al. [16] | $N$ | 1272 | 1090 | 1636 | 262 | 4260 |
| | ±20% | 21 | 46 | 42 | 74 | 39 |
| | ±30% | 38 | 64 | 52 | 85 | 52 |

### 3.3.2. The Modified Correlation and Corresponding Correction Factor

From the above discussions, Watts and Chou's correlation [18] provides the best overall predictive ability. Hence, efforts are made in modifying the correlation by introducing a rational dimensionless parameter, $Gr^*$, into the correlation. Figure 9 shows the correction factor (*CF*) proposed by Watts and Chou [18], $\pi_A$ at wall condition and $Gr^*$ against enthalpy for R-22 [5]. Note that *CF* is defined as:

$$CF = \frac{Nu}{0.021 \times Re^{0.8}\overline{Pr}^{0.55}(\rho_w/\rho_b)^{0.35}}. \tag{12}$$

It was found that the trend of $Gr^*$ is more similar to Nusselt number as compared to $\pi_A$ at the wall condition. $Gr^*$ is more appropriate as the correction factor for heat transfer. Moreover, the threshold proposed Watt and Chou [18] is valid for these organic fluids at lower mass flux. However, it is not applicable at a higher mass flux. Figure 9 shows $Gr^*$ does not reach the threshold value even when HTD happens. The threshold may be revised to the baseline relative to the boundary condition. It is observed that the baseline of $Gr^*$ increase with the heat flux and decrease with the mass flux, implying different resistance that limits HTD happening due to buoyancy force at different conditions. The change of the correction factor, however, almost correspond to the change of $Gr^*$ whatever the heat and mass flux are. The baseline can be estimated using the local or bulk condition of the working fluids:

$$Gr^*_{base} = \frac{\overline{Gr_b}}{Re_b^{2.7}\overline{Pr}_b^{0.5}}, \tag{13}$$

where $\overline{Gr}_{inlet}$ can be written as:

$$\overline{Gr_b} = \frac{g\rho(\rho - \overline{\rho})D^3}{\mu^2} = \frac{(\rho_b - \overline{\rho})}{(\rho_b - \rho_w)}\frac{(\rho_b - \rho_w)}{\rho_b}\frac{gD^3}{\nu^2} \approx \frac{(\rho - \overline{\rho})}{(\rho - \rho_w)}\frac{g\beta\Delta TD^3}{\nu^2}, \tag{14}$$

where $\overline{\rho}$ is the density calculated by integration-averaging with temperature in between the wall and bulk conditions. In the subcritical region, the change of density with the temperature is quite small and $\overline{\rho}$ can be assumed as the arithmetic average density of the wall and bulk. The ratio of the change of density may be written below:

$$\frac{(\rho_b - \overline{\rho})}{(\rho_b - \rho_w)} \approx 0.5. \tag{15}$$

The temperature difference between wall and bulk condition can be written as:

$$\Delta T = \frac{q}{\text{HTC}}. \tag{16}$$

In subcritical region, the buoyancy effect is much smaller than the inertia effect. The forced convection dominates. Hence, the Dittus-Boelter correlation can be applied to estimate the heat transfer coefficient:

$$\text{HTC} = \frac{k_b}{D} \times 0.023 Re_b^{0.8} Pr_b^{0.4}. \tag{17}$$

According to Equations (13–17), $Gr^*_{base}$ can be written as:

$$Gr^*_{base} = \frac{0.5}{0.023} \frac{qg\beta D^4}{k_b \nu^2 Re_b^{3.5} \overline{Pr}_b^{0.9}}. \tag{18}$$

Figure 10 shows the effect of these dimensionless parameters such as $\pi_{A,b}$, $\pi_{A,w}$, $Gr^*$ and $Gr^*/Gr^*_{base}$. The analysis includes six databases that are considered in this study. It shows the deviations of database pertaining to $Gr^*/Gr^*_{base}$ is much better than other dimensionless parameters. As a consequence, $Gr^*/Gr^*_{base}$ is selected as the correction factor for developing the new correlation. Eventually, the derived new correlation is as follows:

$$Nu = 0.0219 \times Re_b^{0.8} \overline{Pr}^{0.55} (\rho_w/\rho_b)^{0.35} (Gr^*/Gr^*_{base})^{-0.58}. \tag{19}$$

The correlation takes the basic form from Watts and Chou [18]. Yet $Gr^*$ is replaced by $Gr^*/Gr^*_{base}$ as the correction factor and the fitting constant is slightly higher changing from 0.021 to 0.0219.

Figure 11 and Table 9 shows the comparison between the predictive Nusselt number and the experimental data all literature. From the table, the proposed correlation offers an average deviation of 0.007 and a standard deviation of 0.181 for all data (4260 in total). With the proposed correlation, Table 10 also indicates that 90% of the R-22 data, 94% of the R-134a data, and 95% of the R-245fa are within the range of ±30%. In essence, the proposed correlation yields the best predictive ability for the organic fluids except ethanol.

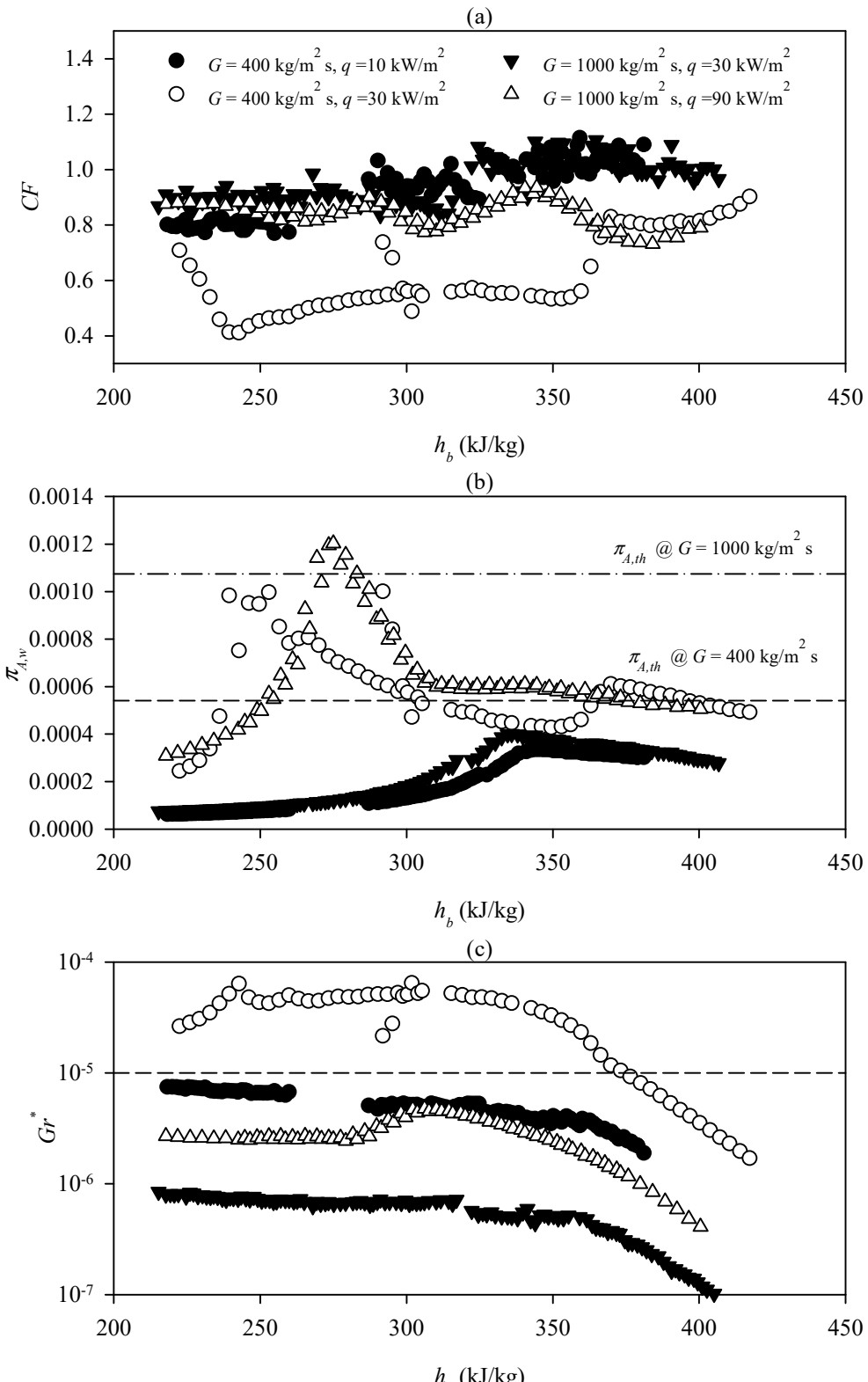

**Figure 9.** (**a**) *CF* proposed by Watts and Chou [18], (**b**) $\pi_A$ at wall condition, and (**c**) *Gr\** against enthalpy for R-22 [5].

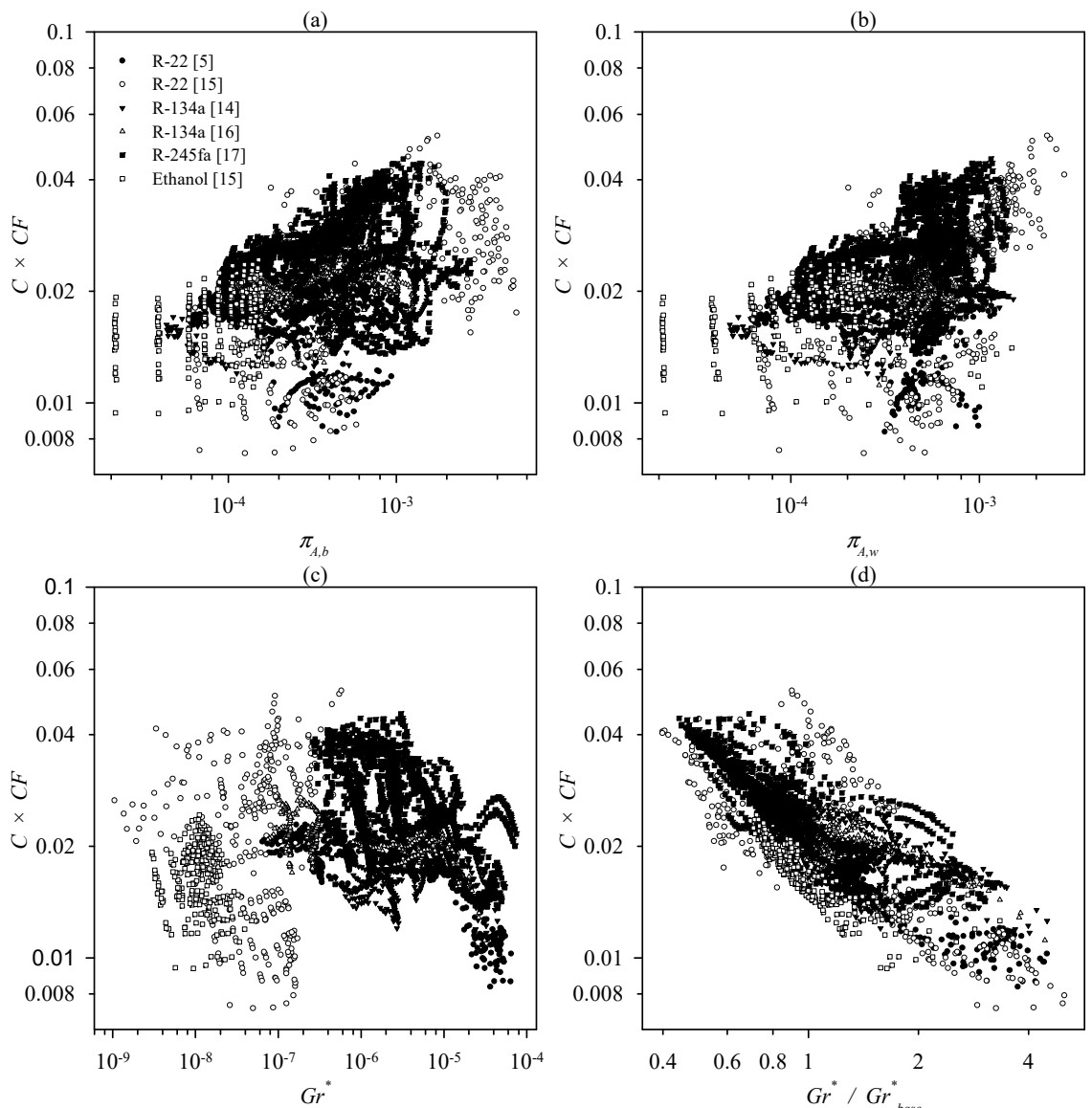

**Figure 10.** The deviation of pertaining to (**a**) $\pi_{A,b}$ , (**b**) $\pi_{A,w}$, (**c**) $Gr^*$, and (**d**) $Gr^*/Gr^*_{base}$.

**Table 9.** Comparisons between the experimental data and the proposed correlation.

| Correlation | | R-22 | R-22 | R-134a | R-134a | R-245fa | Ethanol | Total |
|---|---|---|---|---|---|---|---|---|
| | | Yamashita et al. [5] | Jiang et al. [15] | Kang and Chang [14] | Zhang et al. [16] | He et al. [17] | Jiang et al. [15] | |
| Present study | $N$ | 927 | 345 | 560 | 530 | 1636 | 262 | 4260 |
| | $AD$ | 0.076 | −0.059 | 0.043 | −0.013 | −0.084 | 0.279 | 0.007 |
| | $SD$ | 0.090 | 0.292 | 0.196 | 0.124 | 0.137 | 0.105 | 0.181 |

**Table 10.** The predictive ability of the proposed correlation for heat transfer coefficient of supercritical organic fluids.

| Correlation | | R-22 | R-134a | R-245fa | Ethanol | Total |
|---|---|---|---|---|---|---|
| Presented study | $N$ | 1272 | 1090 | 1636 | 262 | 4260 |
| | ±20% | 82 | 70 | 86 | 16 | 76 |
| | ±30% | 90 | 94 | 95 | 42 | 90 |

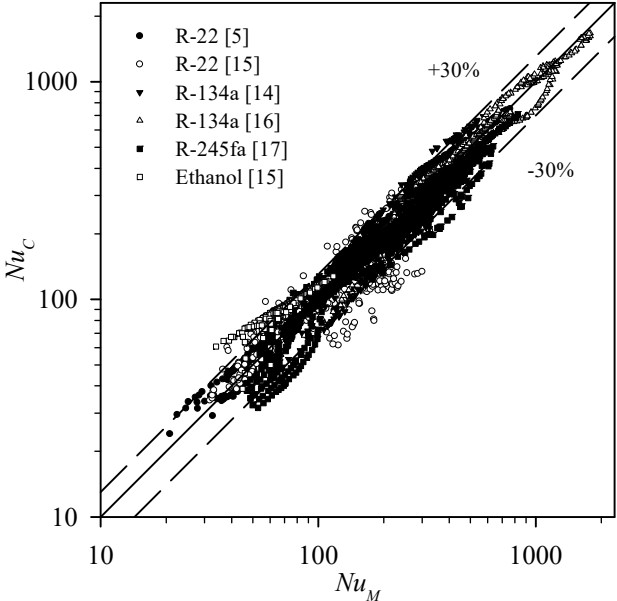

**Figure 11.** Comparison of Nusselt numbers between the proposed correlation and all experimental data.

For ethanol, the study also provides another new correlation to estimate the heat transfer and show as:

$$Nu = 0.0165 \times Re_b^{0.8} \overline{Pr}^{-0.55} (\rho_w/\rho_b)^{0.35} \left(Gr^*/Gr_{base}^*\right)^{-0.8}. \tag{20}$$

Figure 12 shows the new correlation for ethanol that is in good agreement with the experimental data from Jiang et al. [15]. Additionally, 96% of the ethanol data are within ±20% of predictions.

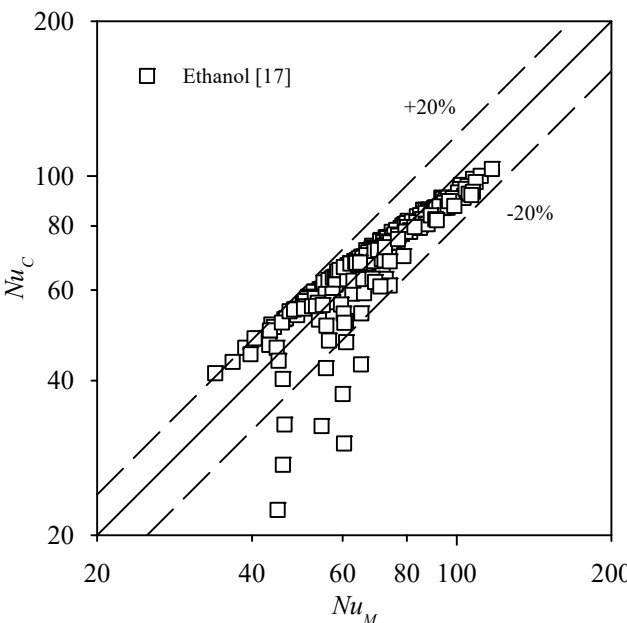

**Figure 12.** The comparison between the measured and calculated Nusselt number for ethanol.

## 4. Conclusions

This study investigated the heat performance of organic fluids for smooth tubes with fluid flowing upward. The organic fluids include R-22, R-134a, R-245fa, and ethanol. It was found that the limit heat flux for organic fluid near the critical point is much smaller than that of the water and carbon dioxide

due to the difference in the ratio of the thermal expansion coefficient and the specific heat capacity. The study presents a new criterion of limit heat flux (LHF) for organic fluids and the proposed criterion of LHF for organic fluid is superior to the existing criteria. The correlation shows an average deviation of -0.232 and a standard deviation of 0.336 for organic fluid.

In the investigation about the onset of the heat transfer deterioration, acceleration parameter imposes a significant effect on the onset of heat transfer deterioration (HTD) due to laminarization. The acceleration parameter, $\pi_A$, imposed appreciable influence on the heat transfer deterioration. For accurate assessment of HTD, the estimation for threshold $\pi_A$ should be at the wall condition rather than at the bulk condition. The threshold $\pi_A$ is strongly related to the limit heat flux and property at the pseudo-critical point.

Based on the evaluations of the existing correlations against existing data, it was found that most correlations are only applicable to their own datasets. Through detailed analysis of the prior data and correlations, the non-dimensional parameter $Gr^*$ offers better predictive ability of the heat transfer coefficient for organic fluids. The correlation takes the basic form from Watts and Chou [18]. Yet, $Gr^*$ is replaced by $Gr^*/Gr^*_{base}$ as the correction factor. The new correlation can provide the smallest average deviation of 0.007 and standard deviation of 0.181 against all data among these common correlations for supercritical conditions. The 90% of 4260 data are within the range of ±30% of the presented correlation. It is the best among other correlations. On the other hand, the study also presents another new correlation for ethanol due to the worse agreement of previously presented correlation, and 96% of the 262 data for ethanol are within the range of ±20% of the correlation.

**Author Contributions:** All the authors have contributed their efforts to complete the paper. Conceptualization, Y.-M.L. and C.-C.W.; Formal analysis, Y.-M.L.; Funding acquisition, J.-S.L. and C.-C.W.; Methodology, Y.-M.L.; Supervision, J.-S.L. and C.-C.W.; Writing—original draft, Y.-M.L.; Writing—review and editing, C.-C.W. All authors have read and agreed to the published version of the manuscript.

**Funding:** This research was funded by the Bureau of Energy, Ministry of Economic Affairs of Taiwan and Ministry of science and technology, Taiwan, grant number 108-2622-E-009-027-CC2 and 108-2221-E-009-037-MY3.

**Acknowledgments:** The authors are indebted to the financial support from the Bureau of Energy, Ministry of Economic Affairs of Taiwan, and grants from Ministry of science and technology, Taiwan.

**Conflicts of Interest:** The authors declare no conflict of interest.

## Nomenclature

| | |
|---|---|
| $A$ | Area, m$^2$ |
| $AD$ | Average deviation |
| $C$ | Correlation constant |
| $CF$ | Correction factor |
| $Cp$ | Specific heat capacity, kJ/kg·K |
| $\overline{Cp}$ | Integrated average specific heat capacity, kJ/kg·K |
| $D$ | Diameter, m |
| $E$ | Eckert number defined by the correlation of Yamagata et al. [8] |
| $F$ | Correction factor defined by the correlation of Zhang et al. [16] |
| $G$ | Mass flux, kg/m$^2$·s |
| $g$ | Gravity acceleration, 9.81 m/s$^2$ |
| $Gr$ | Grashof number |
| $\overline{Gr}$ | Grashof number estimated by integrated average density |
| $Gr^*$ | Buoyancy criterion defined by the correlation of Watts and Chou [18] |
| $h$ | Enthalpy, kJ/kg |
| $k$ | Thermal conductivity, W/m·K |
| $Kv$ | Acceleration factor proposed by McEligot et al. [9] |
| $N$ | Number of data |
| min | Minimum operator |
| $n$ | Exponent of dimensionless factor |

| $Nu$ | Nusselt number |
|---|---|
| $P$ | Pressure, MPa |
| $Pr$ | Prandtl number |
| $\overline{Pr}$ | Integrated average Prandtl number |
| $q$ | Heat flux, kW/m$^2$ |
| $Re$ | Reynolds number |
| $SD$ | Standard deviation |
| $T$ | Temperature, K |
| $u$ | Velocity, m/s |
| $x$ | Unit length of flow direction, m |

**Greek letter**

| $\beta$ | Thermal expansion coefficient, 1/K |
|---|---|
| $\mu$ | Dynamics viscosity, Pa·s |
| $\nu$ | Kinematic viscosity, m$^2$/s |
| $\rho$ | Density, kg/m$^3$ |
| $\overline{\rho}$ | Integrated average density, kg/m$^3$ |
| $\pi_A$ | Acceleration parameter proposed by Cheng et al. [10]. |
| $\xi$ | Fanning fiction factor |

**Subscript**

| $b$ | Bulk condition |
|---|---|
| $base$ | Baseline |
| $C$ | Calculated value |
| $i$ | Index |
| $M$ | Measured value |
| $pc$ | Pseudo-critical condition |
| $th$ | Threshold |
| $w$ | Wall condition |

**Abbreviations**

| HTC | Heat transfer coefficient |
|---|---|
| HTD | Heat transfer deterioration |
| LHF | Limit heat flux |
| ORC | Organic Rankine cycle |

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
