# Peer review of "A Criterion of Heat Transfer Deterioration for Supercritical Organic Fluids Flowing Upward and Its Heat Transfer Correlation"

_energies, doi:10.3390/en13040989_

Round 1
Reviewer 1 Report
In this study, the authors proposed the new supercritical heat transfer correlation applicable for organic fluids when flowing upward in smooth tubes based on the available experimental data. The proposed new correlation can offer the smallest average deviation. The manuscript was well prepared and the topic is of interest to the readership of this Journal. Before deciding publication, there are several issues to be considered.
Please modify the image quality. Please emphasize the practical application of the present study Please emphasize the improvement of the present study compared with the previous studies.
Reviewer 2 Report
Report on "A criterion of heat transfer deterioration for supercritical organic fluids flowing upward and its heat transfer correlation" by Yung-Ming Li, Jane-Sunn Liaw, and Chi-Chuan Wang, submitted to Energies.
The authors present a new correlation for estimating the supercritical heat transfer coefficient for organic fluids based on experimental data. Overall, this is a well-organized manuscript and written in clear language, but a few issues should be addressed to improve the content:
- There are several variables/parameters in Fig. 1 and 2 that first appear in the manuscript and thus should be clearly defined or noted before using them.
- Why is understanding the heat transfer performance of organic fluids important to us compared to inorganic fluids? The motivation behind it was not clearly discussed. Therefore, I would suggest the authors reframe some portions of the introduction and may include more reference articles to highlight the motivation and novelty of the work on organic fluids.
- From Fig. 5, it is questionable to say that the experimental LHF data agrees well with Cheng’s criterion for R-22, given that there were only two data points presented.
- The presentation of Figure 10 and 11 should be improved as it is difficult to acquire accurate information for each fluid from the figures. It might be helpful by making the symbols smaller.
- A few typing and formatting errors need to be checked carefully.
Reviewer 3 Report
The topic of the supercritical organic fluids and its heat transfer correlations is very interesting and strategic for its well known applications. In general it is a good work: the background and the introduction of this paper is very various and complete; the methodology of the work is well described; while the results and its conclusions are well explained.
However it is highly reccomended to improve the paper: by subdividing better the cases analysed also with distinct sub-paragraphs, because it is a bit confusonary in certain places. Moreover it is recommended to improve the Table 6. Finally there are not all the parameter in the nomenclature section.
